# Identification of the Core Pollen-Specific Regulation in the Rice *OsSUT3* Promoter

**DOI:** 10.3390/ijms21061909

**Published:** 2020-03-11

**Authors:** Dandan Li, Rucong Xu, Dong Lv, Chunlong Zhang, Hong Yang, Jianbo Zhang, Jiancheng Wen, Chengyun Li, Xuelin Tan

**Affiliations:** 1State Key Laboratory for Conservation and Utilization of Bio-Resources in Yunnan, Yunnan Agricultural University, Kunming 650201, Yunnan, China; 2Post-Doctoral Research Station of Plant Protection as first class discipline, Yunnan Agricultural University, Kunming 650201, Yunnan, China; 3Rice Research Institute, Yunnan Agricultural University, Kunming 650201, Yunnan, China; 4Yunnan Engineering Research Center for Japonica Hybrid Rice, Kunming 650201, Yunnan, China

**Keywords:** rice, pollen-specificity, expression, *OsSUT3*, promoter

## Abstract

The regulatory mechanisms of pollen development have potential value for applications in agriculture, such as better understanding plant reproductive regularity. Pollen-specific promoters are of vital importance for the ectopic expression of functional genes associated with pollen development in plants. However, there is a limited number of successful applications using pollen-specific promoters in genetic engineering for crop breeding and hybrid generation. Our previous work led to the identification and isolation of the *OsSUT3* promoter from rice. In this study, to analyze the effects of different putative regulatory motifs in the *OsSUT3* promoter, a series of promoter deletions were fused to a *GUS* reporter gene and then stably introduced into rice and *Arabidopsis*. Histochemical GUS analysis of transgenic plants revealed that p385 (from −385 to −1) specifically mediated maximal GUS expression in pollen tissues. The S region (from −385 to −203) was the key region for controlling the pollen-specific expression of a downstream gene. The E1 (−967 to −606), E2 (−202 to −120), and E3 (−119 to −1) regions enhanced ectopic promoter activity to different degrees. Moreover, the p385 promoter could alter the expression pattern of the 35S promoter and improve its activity when they were fused together. In summary, the p385 promoter, a short and high-activity promoter, can function to drive pollen-specific expression of transgenes in monocotyledon and dicotyledon transformation experiments.

## 1. Introduction

Promoters, the upstream sequences of the gene coding sequences, are important molecular biological tools that play a crucial role in the regulation of expression of target genes. This region harbors many cis-elements influencing the expression of a downstream gene, for example, 5′ untranslated region (5′ UTR). In general, constitutive promoters, which drive the expression of target genes uniformly in most parts of a host plant, are commonly used as a means to introduce genes of interest in cereals crops, such as the *CaMV35S* [1], *actin* [2], and maize *ubiquitin* [3] promoters. However, the continuous and efficient expression pattern driven by these is usually unnecessary and results in a large amount of nutrition consumption in the plant body, which negatively affects plant growth and development [4,5,6]. Therefore, it is particularly urgent to exploit tissue-specific (space) or time-specific (temporal) promoters for bio-fortifying target gene products restrictedly to only specific parts or time periods in a host plant in order to avoid a yield penalty from ubiquitous target gene expression.

In flowering plants, the proper development of viable pollen is crucial for successful sexual reproduction and maintaining the continuity of a species. As an important cereal crop, rice (*Oryza sativa*) has been shown to require well-developed pollen for obtaining a high yield, utilizing heterosis, and breeding new varieties. At present, a large number of modern biotechnological studies have focused on the molecular mechanisms of pollen-specific gene regulation to increase pollen activity and viability [7,8,9,10]. Pollen-specific promoters are extremely useful tools for studying pollen development by molecular genetics and have quickened molecular breeding with male sterile lines [11].

To date, many promoters have been identified as pollen-specific in several species, but strong pollen-specific promoters that can be used for research in various crops are limited [12]. The maize *Zm13* gene was first described to be expressed specifically in pollen [13,14], and its regulatory region was shown to activate expression during microgametophyte development [15]. In addition, the promoters of the late pollen-specific family (Latency-associated transcript, LAT) from *Lycopersicum esculentum* have been well-studied and are the best characterized sequences regulating the pollen-specific expression of genes [16,17]. Some cis-acting elements from the sequences of promoters regulating pollen-specific gene expression have been recently identified and were found to be highly-conserved across multiple species [6]. The PB core motif (GTGA), the POLLEN1LeLA52 motif (AGAAA) and the “TCCACCATA” motif were all identified in the *LAT52* pollen-specific promoter in tomato [18,19,20]. The GTGANTG10 motif (CCAC) in the tobacco *g10* gene promoter was demonstrated to play an important role in the regulation of pollen-specific gene expression [21]. Moreover, enhancer motifs in the promoter region have been found to be essential for high levels of tissue-specific gene expression, such as hTERT and CMV elements [22], the photoreceptor regulatory element-1 [23], and the C-repeat/dehydration-responsive element [24]. Thus, detailed research on pollen-specific promoters should further our understanding of the molecular regulation mechanism of pollen development and sexual reproduction.

Compared to other crop plants, fewer promoters that control pollen-specific expression have been identified in rice, limited to promoters such as *Osg6B*, *OsIPP3*, *OsCPK21*, *OsIPP3*, *OsIPA*, *OsRA8*, *OsMADS63* and *OsMADS68* [6,25,26,27,28,29]. Among these, many promoters could not maintain strong and specific expression in pollen when they were transported into a heterologous system, such as *Arabidopsis* [30]. Thus, such promoters are limited and problematic for routine use as a regulatory tool in most cereal transformation initiatives.

*OsSUT3*, a member of the sucrose transporter (*SUT*) gene family in rice, is highly expressed in developing pollen and may play a major role in transporting sucrose into pollen for starch accumulation [31,32]. The isolation and characterization of the *OsSUT3* gene started more than a decade ago, but to date, its regulatory region and specific function in pollen development has not been studied. In a previous study, we isolated the *OsSUT3* gene promoter for the first time, providing a convenient tool for the research of promoter function [33].

In this study, we further identified the core regulatory regions in the pollen-specific promoter of the *OsSUT3* gene by fusion with the *GUS* reporter gene. The activities of different regulatory regions of the *OsSUT3* promoter were evaluated both in rice (a monocot model plant) and *Arabidopsis* (a dicot model plant). Based on molecular and biochemical analyses of the transformants, we characterized a key region for pollen-specific control of gene expression.

## 2. Results

### 2.1. Pollen-Specific Motifs in the OsSUT3 Promoter and the Deletion

In our previous study, the promoter of the *OsSUT3* gene was cloned and was determined to be 2029 bp [33]. In the present study, this version of the promoter was termed “pOsSUT3” (Figure 1). Motif analysis of pollen-specific elements in this region revealed three different types of cis-elements that control pollen-specific expression, which were respectively POLLEN1LeLA52, PB CORE, and GTGANTG10. Six POLLEN1LeLA52 sites (AGAAA, at −1387, −1301, −1252, −1037, −302, and −286), seven PB CORE sites (GTGA, at −1598, −1006, −865, −606, −348, −242, and −154), and ten GTGANTG10 sites (CCAC, at −1572, −1399, −1345, −1017, −900, −852, −807, −732, −469, +347, and +428) were identified in the *OsSUT3* promoter (Figure 1).

Based on the location of POLLEN1LeLA52, five specific primers were designed (Table 1), and a full 5′ UTR sequence was synthesized, obtaining six promoter variants that we termed p1016, p847, p486, p426, p385, and p746 (Figure 1). pOsSUT3 was further divided into eight parts, P1–P8, according to the intersection and overlap of the promoter fragments above (Figure 1). The pollen-specific motifs that fell within every part are summarized in Table 2.

### 2.2. Identification of the Core Pollen-Specific Region in the OsSUT3 Promoter

Next, six promoter fragments were independently fused to the *GUS* reporter gene and used in the *Agrobacterium*-mediated genetic transformation of rice (Figure 2A). We chose transformed rice plants with the constitutive CaMV35S promoter-*GUS* vector and non-carrier plants as references for comparative analysis. To determine the localization of the product of the *GUS* reporter gene under the control of variants of the *OsSUT3* promoter, several organs and tissues were analyzed using histochemistry, including the root, stem, leaf, and panicle (Figure 2B). In *Nipponbare* untransformed plants, no *GUS* activity was observed in discolored tissues. The *GUS* activity under the control of the 35S promoter was high and relatively coincident in every tissue. However, in panicles, the *GUS* activity driven by the 35S promoter was observed only in the paddy hull and not in the pollen. Stained plant lines carrying different promoter fragments showed that the p1016, p847, p486, p426, and p385 fragments were all able to drive pollen-specific expression of *GUS* in rice, but the p746 insert lacked tissue-specificity and drove *GUS* gene expression in all tested tissues, thus serving as a constitutive promoter. Among the five pollen-specific promoter fragments, the *GUS* activities induced by p847and p385 were higher in pollen than those of the p1016, p486, and p426 fragments. These results indicated that two regions in the *OsSUT3* promoter were sufficient to highly express a reporter gene in a pollen-specific manner.

To determine the strength of the six *OsSUT3* promoter variants, the quantitative expressions of *GUS* mRNA in leaves, stems, panicles, and roots were determined (Figure 2C). Consistent with the histochemical staining of *GUS*, the levels of *GUS* transcript could be designated as “no expression” in untransformed rice, and high and relatively balanced *GUS* expression was detected in various tissues of lines transformed with CaMV35S::*GUS*. Further, the *GUS* expression in other independent transgenic lines, except for the p746 insert line, exhibited significant increases and distinct specificity in pollen compared with that in the leaf, stem, and root. These results also confirmed that p1016, p847, p486, p426, and p385 all had regulatory sequences necessary for downstream pollen-specific gene expression. Compared to p746, which had no tissue-specific expression in pollen and was relatively long, p385, the shortest pollen-specific promoter fragment, only contained 182 bp of the 5′-region from −203 to −385 (Figure 3. However, this 182-bp sequence led to noticeably specific expression of a reporter gene in pollen. Therefore, this region (from −203 to −385) was confirmed to be a core pollen-specific region of the *OsSUT3* promoter.

### 2.3. Identification Three Enhancer Regions (E1, E2, and E3) That Control GUS Expression in pOsSUT3

*GUS* gene expressions driven by various promoter variants exhibited different levels in the same tissue, for example, in the panicle (Figure 2B,C). Based on the expression level of *GUS* mRNA and the location of the promoter fragments, we identified three enhancer regions in the pOsSUT3 sequence, which we named Enhancer Region 1 (E1, 362 bp, from −967 to −606), Enhancer Region 2 (E2, 84 bp, from −203 to −120), and Enhancer Region 3 (E3, 119 bp, from −119 to −1) from the 5′ side (Figure 3). Our previous findings showed that the *GUS* expression driven by pOsSUT3 harboring E1, E2, and E3 was higher than that in any other promoter fragments and was 10.75-times the expression controlled by the 35S promoter in the panicle. In this study, the highest *GUS* expression in the panicle was detected in plants harboring p385::*GUS*; this expression increased approximately 8.27-fold compared with that in plants carrying 35S::*GUS* (Figure 2C). The second-highest *GUS* expression in the panicle was found in transgenic plants induced by p746::*GUS,* though this was not specific to any tissue, despite being 8.49-fold as high as that in 35S::*GUS* lines (Figure 2C). However, compared with pOsSUT3, there was a 13.76% and 21.04% reduction in the panicle *GUS* expression driven by p385 and p746, respectively. The absence of E3 in p847, which was 2.18 times the expression of 35S::*GUS* lines, resulted in a significant reduction (79.72%). When we removed E1 and E3 together (p486), the *GUS* expression was only 21.57% of 35S::*GUS* lines. The removal of E2 and E3 at the same time, for instance, in construct p1016, drastically reduced the *GUS* expression to 18.66% of 35S::*GUS* plants. Finally, when we deleted all three enhancer regions with only a small part of E2, the *GUS* expression reached a minimum that was 5.72% of 35S::*GUS* plants. Taken together, these results suggested that the deletion of E1 relatively decreased *GUS* expression, but a higher expression could still be maintained by E2 and E3. E2 had the greatest enhancement effect on reporter gene expression, followed by E3. The three regions showed the additive up-regulation effect on a downstream gene.

Comprehensive sequence analyses of enhancer motifs were carried out in E1, E2, and E3 regions (Table 3). These results revealed core binding sites (ACGT and AAAG) that control gene expression and cis-acting enhancers. E1 had three cis-acting elements, which respectively were one AAAG motif and two CAAT-boxes, while E2 had one CAAT-box and E3 had one ACGT motif. These factors might play an important role in mediating the high-level expression of a downstream gene.

### 2.4. p385 Can Drive Pollen-Specific Expression in the Dicotyledonous Model Plant Arabidopsis thaliana

Our *OsSUT3* promoter fragment activity analysis suggested that the regulatory regions required for pollen-specificity and high expression were located in the p385 fragment identified by rice transgenic engineering. To test whether p385 was able to drive reporter gene expression specifically in the pollen of other species, we fused this fragment to the *GUS* gene and introduced a *p385::GUS* vector into the dicotyledonous model plant *Arabidopsis* (Figure 4A). Histochemical *GUS* analysis on a wide range of tissues from transgenic plants carrying p385::*GUS* showed that the p385 fragment could drive pollen-specific expression, with little to no expression detectable in other tissues examined compared with lines carrying 35S::*GUS* (Figure 4B). Further analysis by quantitative real-time polymerase chain reaction (qRT-PCR) demonstrated consistent results as compared to *GUS* lines (Figure 4C). However, the *GUS* expression in the panicles of p385::*GUS* lines, 52.19% of the level of that of 35S::*GUS* plants, was significantly weaker. These results indicated that the p385 region of the *OsSUT3* promoter was effective for expressing a downstream gene in a pollen-specific manner, despite the gene expression level being observably decreased compared to the level as a gene driven by the 35S promoter.

### 2.5. p385 Alters the Expression Pattern of the CaMV35S Promoter in Rice

The *GUS* expression driven by the p385 fragment in *Arabidopsis* pollen was lower than that driven by 35S; thus, we hypothesized that gene expression would increase when a strong promoter was fused in front of p385. A reporter construct fusing two promoters (35S and p385) to the *GUS* gene was developed and used to generate transgenic rice lines (Figure 5A). Histochemical analysis of different tissues showed that p385 from the *OsSUT3* promoter changed the 35S expression pattern and expressed the reporter gene only in rice pollen. A quantitative evaluation of *GUS* transcript levels also suggested that the *GUS* expression induced by both 35S and two-promoters were consistent with previously observed *GUS* signaling levels (Figure 5B). Compared with the wide *GUS* expression driven by only the 35S promoter, prominent and higher pollen-specific expression (2.15-fold) was detected in transgenic lines carrying both 35S and p385. These results demonstrated that the p385 promoter might play an important role in the regulation of gene expression location in plants, and an enhancement effect of gene expression also existed.

## 3. Discussion

### 3.1. p385 Is the Core Pollen-Specific Regulatory Region of the OsSUT3 Promoter

As an important means for driving gene expression, several exogenous promoters have been widely used for genetic engineering in plants. However, it has been demonstrated that the constitutive expression of genes may cause energy consumption, metabolic disturbance, and some unexpected traits [37,38]. For more precisely targeted studies of gene function, the use of tissue-specific promoters that harbor specific motifs for controlling exogenous gene expression in specific organs and tissues have many advantages [39]. Similar to other tissue-specific promoters, the core promoter of *OsSUT3* includes the minimal region of the DNA sequence, including a TATA-box, initiator, and tissue-specific regulatory elements [40]. The TATA-box has been verified to positively regulate the expression of the Sea urchin *H2A* gene, and the deletion of this region reduces *H2A* expression by 15 to 20-fold [41,42]. Pollen-specific activation elements (AGAAA, GTGA, and CCAC) were found to be essential for gene pollen-specific expression [17,20,21]. These cis-acting regulatory elements were all detected in the *OsSUT3* promoter region (Table 3, Figure 1). To use the *OsSUT3* promoter more efficiently and conveniently in transgenic plants, the aim of this study was to clarify which region was the core sequence in this promoter for the purpose of regulating gene pollen-specific expression.

Rice transformants, p1016, p847, p486, p426, p385, and p746, with different regions of the *OsSUT3* promoter showed varied levels of *GUS* pollen-specific expression. Histochemical and quantitative analyses of *GUS* expression all showed that p847 and p385 promoted high-level and pollen-specific reporter gene expression; p385 was the minimal fragment in all versions (Figure 2B,C). Between the two promoters, p385 not only drove the highest expression of *GUS* in pollen but also revealed a small significant expression in leaf. However, p847 also drove a relatively good expression in pollen but with no apparent leakage in other tissues. We speculate that the 119-bp region (from −119 to −1) next to ATG contained some cis-elements that could regulate downstream gene expression in leaves. Similar to our finding, *GUS* pollen-specific activity was not detected in rice transformants carrying p746, which removed the 5′ 183-bp region from p385 (from −385 to −203) (Figure 3). However, further research is needed to confirm this point. In addition, bioinformatic analysis of pollen-specific elements showed there were four key regulators, two POLLEN1LeLA52 elements (AGAAA), and two PB CORE elements (GTGA) located in this region. Therefore, this region was the core pollen-specific fragment of the *OsSUT3* promoter and played a critical role in the determination of promoter strength and its specific expression.

However, the promoter activity was weaker than expected in heterologous systems, and even exhibited non-specificity at times [43,44]. Many reports have suggested that most regulatory functions of cis-acting elements between monocots and dicots are different. For instance, the most efficient promoters in tobacco cells show significant differences from the gene specificity expression in rice cells [45]. In this study, the p385 promoter activity was analyzed in *Arabidopsis* to investigate the function of this promoter in various species (Figure 4B). These results revealed that the p385 promoter could drive high *GUS* gene expression specifically in *Arabidopsis* pollen, which suggests p385 has the same pollen-specific activity in a heterologous system.

An overview of the results in transgenic rice and *Arabidopsis* indicated that p385 facilitates high reporter gene activity in the pollen of monocots and dicots. The core pollen-specific regulatory motif was present in the 183 bp region located in the 5′ region of p385.

### 3.2. p385 Was Characterized by Two Gene Expression Motifs

In most studies, it is a widely used strategy to functionally dissect the parts of a gene promoter for the analysis of regulatory regions [46,47]. Although the p385 promoter containing a minimal 183-bp fragment was demonstrated to control pollen-specific expression, the other 1846-bp region carrying a large amount of cis-regulators was required for conferring high-level expression in pollen. Various cis-acting regulatory elements, such as ACGT, AAAG, and CAAT-box enhancer elements involved in pollen-specific expression, were detected in the *OsSUT3* promoter region (Table 2). The ACGT and AAAG core binding sites, known to enhance gene expression in many genes (e.g., *AtGEX2*, *OsGEX2*, and *PtGEX2*) [48,49] were verified to act as target sequences for bZIP DNA-binding regulatory proteins [36,50] and for Dof transcription factors [51], respectively. Moreover, the CAAT-box is a common cis-acting element in the enhancer region of promoters [52].

In this investigation, minor variations in the *OsSUT3* promoter were found to drive the different levels expression in rice pollen (Figure 2B,C), which suggested that deletions in promoters contained some motifs to enhance reporter gene expression. Histochemical and qRT-PCR analyses of panicles from transgenic rice demonstrated that there were three enhancing regions in the *OsSUT3* promoter: E1, E2, and E3 (Figure 3). Maximum *GUS* expression was observed in the panicle of p385 and p746 rice transformants, whereas p1016 and p426 showed negligible *GUS* expression. The main reason for the large difference in *GUS* expression between the two groups was likely the presence of E2 and E3 enhancer elements. Additionally, the E1 region caused higher expression of the *GUS* gene in p847 transformants compared to p486 transgenic plants, which suggests that E1 might be attributed to the high-level expression of a downstream gene. This was consistent with previous results, where different regions of promoters have been used to transcribe the downstream gene at variable strengths depending on the specific cell or tissue [53].

By contrast, the presence of cis-acting enhancers in these regions also demonstrates a potential interaction between the cis-elements present within these three regions and other regulators. However, these results do not exclude that some other enhancers that have not been reported previously are located in these regions.

### 3.3. p385 Changes the Expression Pattern of the Constitutive Promoter CaMV35S

In order to further exploit the usage pattern of p385 from the *OsSUT3* promoter, we used a two-promoter vector-mediated *GUS* expression system and analyzed the expression sites and quantitative traits of the resultant transgenic *GUS* gene (Figure 5). We found that p385 adjacent to the 35S promoter worked as a directed promoter with pollen specificity. p385 altered the 35S promoter characteristics, which constitutively expressed the *GUS* gene in all tested organs or tissues and specifically guided the promoter activity only in pollen tissue. These results suggest that the p385 fragment has cis-acting regulatory elements that were indispensable for robust and pollen-specific promote activity. Additionally, the *GUS* expression of rice panicles with two-promoters was much higher than those with only the 35S promoter.

In most studies, double 35S promoters are widely used to drastically enhance the expression of exogenous target genes [54,55], suggesting that the connection of two strong promoters shows additive effects for driving activity. In this study, we first connected two different types of promoters in front of a reporter gene and confirmed that p385 located adjacent to the *GUS* gene might play a major role in driving the expression pattern in plants. Further, the 35S promoter enhanced the activity of the p385 promoter for expressing a gene at a higher level.

In conclusion, p385 was the strongest driver of six tested promoters to determinate the pollen-specific expression of target genes. In addition, p385 also had been proved to have good stability and high activity of pollen-specific expression in heterologous systems. Therefore, it was a potential tool in agriculture, especially in male-sterility. For instance, the transgenic male sterile lines and maintainer lines can be generated by introducing a pollen-killer gene, fertility-restoration gene. This method is suitable for most flowering plants for breeding new varieties and increasing production of hybrid seeds.

## 4. Materials and Methods

### 4.1. Sequence Analysis and Promoter Deletion

In order to investigate the pollen-specific cis-acting elements in the *OsSUT3* promoter, the full-length promoter sequence cloned in a previous study [33] was used to perform motif alignment with PlantCARE. To verify the function of each part, a total of five promoter deletion fragments were PCR amplified, named p1016 (−1196/−180), p847 (−967/−120), p486 (−606/−120), p426 (−606/−180), and p746 (−203/+543), as well as one promoter deletion fragment, p385 (−385/−1, 5′ UTR of *OsSUT3* gene). The sequences of the specific primers used to amplify fragments are given in Table 1.

### 4.2. Plasmid Construction and Transformation

Subsequently, promoter deletion fragments were fused with the *GUS* reporter gene in the *pBI121* vector (KIB, CAS, Kunming, Yunnan, China) using *Hin*d III and *Nco* I restriction sites. The recombinant constructs were mobilized into *A. tumefaciens* (strain GV3101) using a protocol described in earlier studies [56].

### 4.3. Plant Materials and Growth Conditions

For rice transformants, seeds of the japonica rice cultivar *Nipponbare*, obtained from the Rice Research Institute of Yunnan Agricultural University (Kunming, China), were surface sterilized with a 0.1% HgCl_2_ solution, followed by washing with autoclaved water that was used for genetic transformation. Calli derived from the mature embryos were infected with Agrobacterium culture, and putatively transformed calli were selected on the Murashige and Skoog (MS) medium [57] containing 100 mg/L kanamycin (Kan). Plants regenerated from the selected Kan-tolerant calli were grown to maturity for 6–8 weeks at 26 ± 2 °C under a 16/8 h light/dark cycle in a greenhouse.

For *Arabidopsis* transformants, wild type *Arabidopsis thaliana* (Columbia) plants (KIB, China) were infected with *Agrobacterium* (GV3101) harboring various recombinant vectors carrying different expression units of the *OsSUT3* promoter, respectively, using the floral dip method [58]. Next, all plants were grown for 3–4 weeks at 23 °C under a 16/8 h light/dark cycle.

### 4.4. Histochemical Localization of β-Glucuronidase (GUS) Activity

Histochemical analysis of GUS activity was carried out essentially as described by Jefferson et al. [59]. T1 and T2 plants of rice and *Arabidopsis* transformants were grown to 1–3 days before flowering under controlled conditions. The roots, stems, leaves, and panicles of transgenic lines, as well as control rice plants, were collected and tested for *GUS* expression. All the explants were incubated at 37 °C for 24 h in a staining buffer containing 1 mM 5-bromo-4-chloro-3-indolyl-β-D-glucuronide (X-gluc), 100 mM sodium phosphate buffer (pH 7.0), 2 mM potassium ferrocyanide and potassium ferrocyanide, and 0.1% Triton X-100. Next, the explants were rinsed 3–5 times with 70% ethanol in an 80 °C water bath and then observed under a microscope.

### 4.5. Genome DNA and Total RNA Extraction

Total genomic DNA was isolated from rice leaf tissue using the CTAB method for PCR amplification [60]. All tissues (root, stem, leaf, panicle) prepared for RNA extraction were ground into a fine powder using liquid nitrogen and mini pestles in a 2.0 mL Eppendorf tube. Total RNA was isolated from different tissues (50–100 mg fresh weight) using TRIzol (TIANGEN, Beijing, China) following the manufacturer’s instructions. Next, 1 μL of DNAse was added to the RNA solution (30–50 µL) for eliminating DNA contamination. The quality of RNA samples was by checked using a 1.2% denaturing agarose gel, and the quantity and concentration were estimated using a NanoDrop 1000 (Thermo Scientific Inc., Waltham, MA, USA) spectrophotometer, respectively. First-strand cDNA was synthesized by adding an equal quantity of RNA using the FastKing RT Kit (With gDNase) (TIANGEN) for qRT-PCR.

### 4.6. qRT-PCR

qRT-PCR was performed using a CFX96 Real Time System (Bio-Rad, Hercules, CA, USA) and SuperReal PreMix Plus (SYBR Green) (TIANGEN, Beijing, China) following the manufacturer’s instructions. The β-actin gene was used as an endogenous control. PCR reactions were assembled in a reaction volume of 20 μL containing 10 μL (2×) SYBR green, 0.6 μL (12 μM) of each primer, and 1 μL of 40–50 ng of cDNA. PCR amplification was carried out using the following protocol: UDG activation at 50 °C for 2 min, then held at 95 °C for 5 min, followed by 40 cycles of 95 °C for 5 s and 61 °C for 30 s. Melt curve analysis was performed afterward using default settings. Three independent biological replicates were analyzed. Relative quantification values were calculated using lg2^−ΔΔCt^; the 2^−ΔΔCt^ method can be reviewed in [61].

## 5. Conclusions

In this study, we identified a 183 bp core region from the 2029 bp *OsSUT3* promoter that regulates the pollen-specific expression of a downstream gene in homogenous or heterologous plants. Future work should investigate if the E1, E2, and E3 modules from the *OsSUT3* promoter regulate gene expression at a higher level. Our study also indicated that the p385 promoter containing the 183-bp core region and E2 and E3 enhanced regions could alter the expression pattern of the 35S constitutive promoter and highly express a reporter gene only in pollen. The present study demonstrates that p385 may be exploited as a potential candidate for genetic engineering and the generation of male-sterile lines in higher plants.

## Figures and Tables

**Figure 1 ijms-21-01909-f001:**
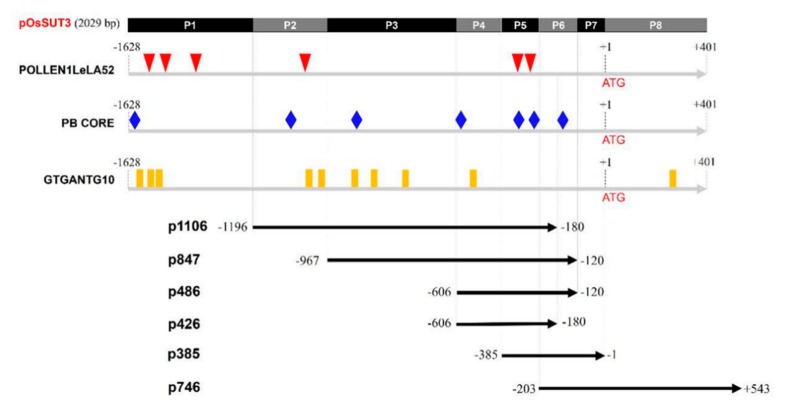
Schematic representation and pollen-specific motifs of various pOsSUT3 deletion constructs (p1016, p847, p486, p426, p385, and p746) used to transform rice. The numbers on the left indicate the upstream end of the promoter fragment present in each construct. The downstream end was on the right side. The top rectangle indicates the full length *OsSUT3* promoter named pOsSUT3. P1–P8 represents the eight parts from pOsSUT3 divided according to pollen-specific elements. Different colored symbols represent the position of three pollen-specific elements, POLLEN1LeLA52, PB CORE, and GTGANTG10.

**Figure 2 ijms-21-01909-f002:**
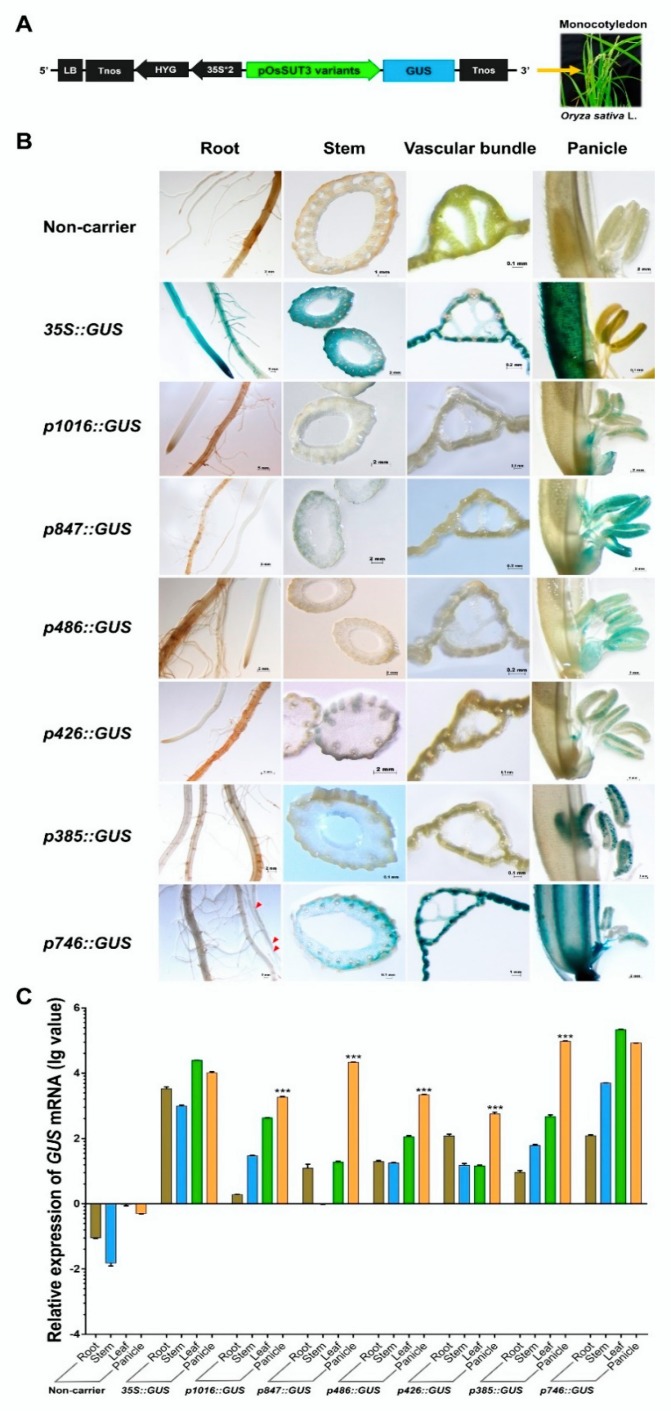
*GUS* expression in various rice transformants and non-carrier plants. (**A**) Linear map of different pOsSUT3 variants for transgenic rice. (**B**) Histochemical localization of GUS activity in vegetative tissues from stably transformed rice plants. The columns from left to right indicate the root, stem, leaf, and panicle. The lines from top to bottom indicate the tissues from non-carrier, 35S, p1016, p847, p486, p426, p358, and p746 lines. The red arrows of p746 staining root indicate GUS activity. (**C**) The relative *GUS* transcript expression assayed by qRT-PCR in the root, stem, leaf, and panicle harvested from p1016, p847, p486, p426, p358, and p746 rice transformants compared to non-carrier plants and 35S::*GUS* transgenic lines. Brown bar chart: root; blue bar chart: stem; green bar chart: leaf; orange bar chart: panicle. Data are shown as the mean ± SD (*n* = 3), student’s *t*-test. *** represents highly significant differences from all tested tissues from each transgenic plant. Relative expression was calculated as: lg2^−ΔΔCt^.

**Figure 3 ijms-21-01909-f003:**
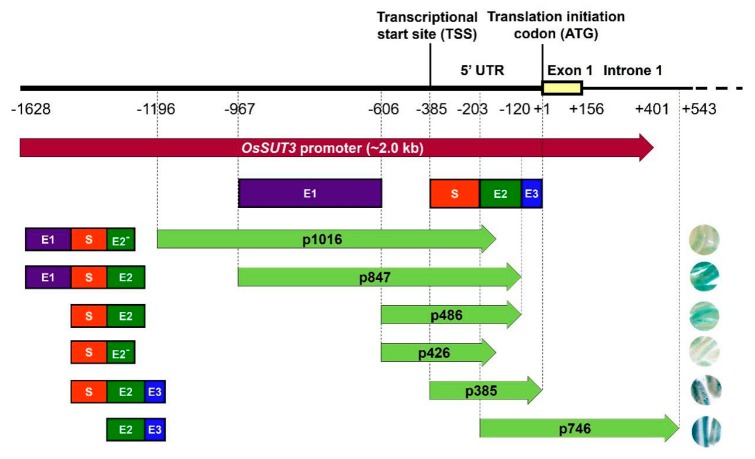
Contribution of four regions from pOsSUT3 to the expression of the *GUS* reporter gene in pollen. Purple rectangle, enhancer region 1 (E1); dark green rectangle, enhancer region 2 (E2); blue rectangle, enhancer region 3 (E3); red rectangle, specific expression region (S). E2^−^ represents the 5′ part of the E2 deleted 60 bp at the 3′ end. The frameworks of rectangular structures on the left of the arrowed rectangle represent the functional regions contained in various promoter deletions. The right circles represent the staining strength of GUS in pollen from transformant lines carrying different promoter fragments.

**Figure 4 ijms-21-01909-f004:**
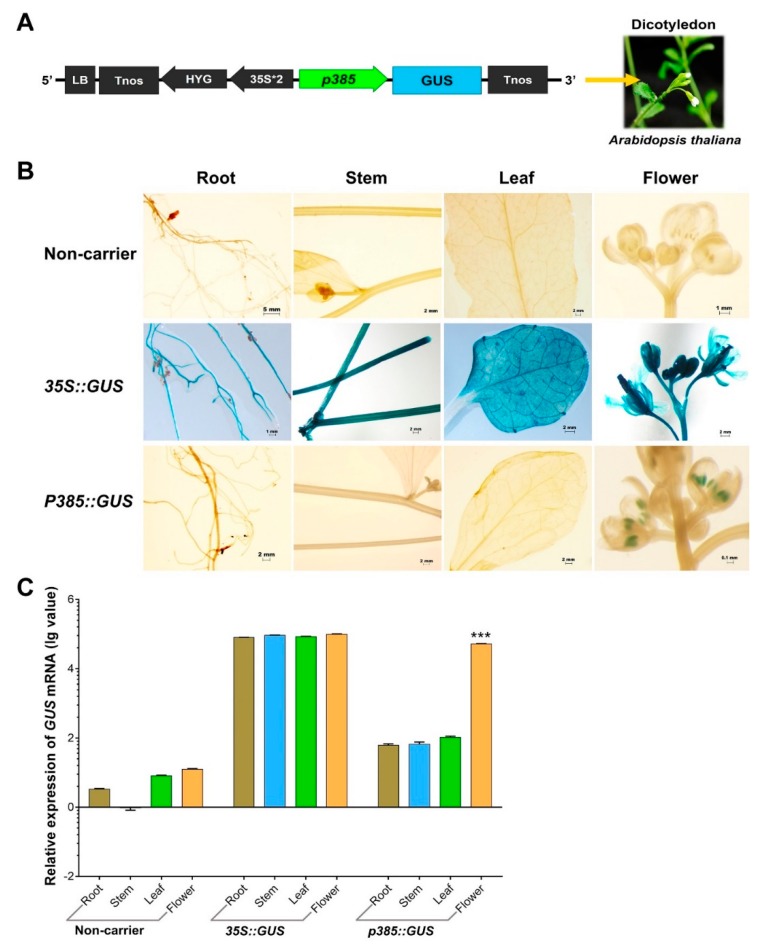
*GUS* expression in p385::*GUS* rice transformants in contrast to non-carrier plants and 35S::*GUS* lines. (**A**) Linear map of the p385::*GUS* vector for transgenic *Arabidopsis*. (**B**) Histochemical localization of GUS activity in vegetative tissues from stably transformed rice plants. The columns from left to right indicate the root, stem, leaf, and panicle. The lines from top to bottom indicate the tissues from non-carrier, 35S, and p385 lines. (**C**) The relative *GUS* transcript expression assayed by qRT-PCR in the root, stem, leaf, and panicle tissues harvested from p358 rice transformants compared to non-carrier plants and 35S::*GUS* transgenic lines. Brown bar chart: root; blue bar chart: stem; green bar chart: leaf; orange bar chart: panicle. Data are shown as mean ± SD (*n* = 3), student’s *t*-test. *** represents highly significant differences from all tested tissues of each transgenic plant. Relative expression was calculated as: lg2^−ΔΔCt^.

**Figure 5 ijms-21-01909-f005:**
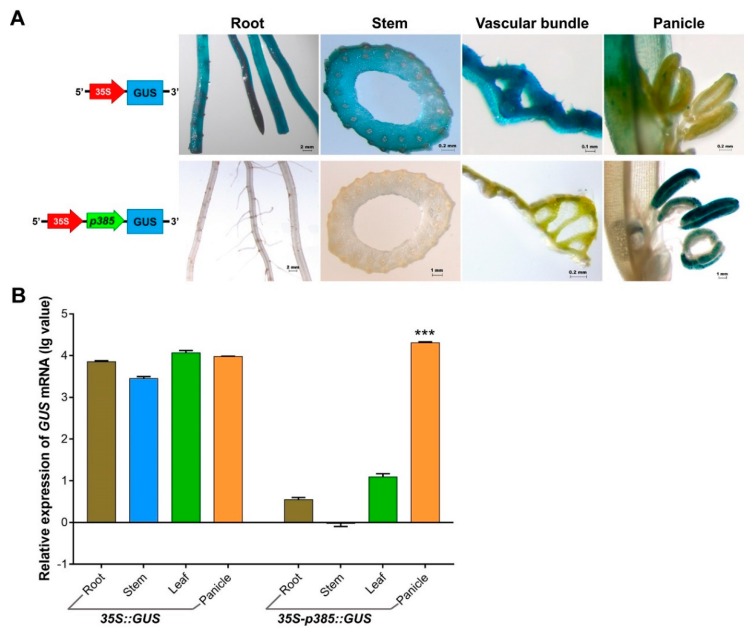
*GUS* expression in various rice transformants as contrasted with non-carrier plants. (**A**) Histochemical analysis of GUS expression in different tissues from non-carrier and transgenic rice plants harboring 35S-p385::*GUS* and p385::*GUS* constructs. Linear maps for transgenic rice are located on the corresponding left of stained tissues. (**B**) Quantitative qRT-PCR analysis of *GUS* transcripts from the root, stem, leaf, and panicle of non-carrier plant and transgenic rice plants harboring 35S-p385::*GUS* and p385::*GUS* constructs. Brown bar chart: root; blue bar chart: stem; green bar chart: leaf; orange bar chart: panicle. Data are shown as mean ± SD (*n* = 3), student’s *t*-test. *** indicates highly significant differences from all tested tissues for each transgenic plant. Relative expression was calculated as: lg2^−ΔΔCt^.

**Table 1 ijms-21-01909-t001:** Primer sequences used in this study.

Primer Name	Sequences (from 5′ to 3′) *	Application
p1016-F	CCCAAGCTTGGGGGTATGTTATAAGGGTCTGGTAGGA	Promoter cloning
p1016-R	CATGCCATGGCATGAGAGGAGGGAGCGGTGAGA	Promoter cloning
p847-F	CCCAAGCTTGGGCCTTCGTATGTAAGGGACGCT	Promoter cloning
p847-R	CATGCCATGGCATGGACCAAGACGACGACGGATA	Promoter cloning
p486-F	CCCAAGCTTGGGCCGCAATGAACTCTGCCTAT	Promoter cloning
p486-R	CATGCCATGGCATGGACCAAGACGACGACGGATA	Promoter cloning
p426-F	CCCAAGCTTGGGCCGCAATGAACTCTGCCTAT	Promoter cloning
p426-R	CATGCCATGGCATGGAGGAGGGAGCGGTGAGA	Promoter cloning
p746-F	CCCAAGCTTGGGCAATGTCTCACCGCTCCC	Promoter cloning
p746-R	CATGCCATGGCATGGCAAACAGGCAGCATAAGAG	Promoter cloning
GUS-F	ATCCTCTGGGAACCACTGAACC	Real-time RT-PCR
GUS-R	CATCACATTGCTCGCTTCGTTA	Real-time RT-PCR
Actin-F	ATCCTTGTATGCTAGCGGTCGA	Real-time RT-PCR
Actin-R	ATCCAACCGGAGGATAGCATG	Real-time RT-PCR

* Underlined letters represent the sites recognized by *Hin*d III and *Nco* I.

**Table 2 ijms-21-01909-t002:** Pollen-specific motif analysis of the pOsSUT3 promoter.

Pollen-Specific cis Element	Sequence Motif	No. in P1	No. in P2	No. in P3	No. in P4	No. in P5	No. in P6	No. in P7	No. in P8	References
POLLEN1LeLA52	AGAAA	3	1	0	0	2	0	0	0	[18]
PB CORE	GTGA	1	1	1	1	2	1	0	0	[16]
GTGANTG10	CCAC	3	2	3	1	0	0	0	1	[21]

**Table 3 ijms-21-01909-t003:** Enhancer motif analysis of E1, E2, and E3.

Enhancer cis Element	Sequence Motif	No. in E1	No. in E2	No. in E3	References
ACGT MOTIF	ACGT	0	0	1	[34]
Prolamin box (P-box)	AAAG	1	0	0	[35]
CAAT box	CAAT/CAAAT/CCAAT	2	1	0	[36]

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
