# Peer review of "Identification of the Core Pollen-Specific Regulation in the Rice OsSUT3 Promoter"

_ijms, 2020, doi:10.3390/ijms21061909_

Round 1
Reviewer 1 Report
In this work, the authors characterize a rice promoter to drive gene expression specifically in pollen. For that, they construct and analyze partial deletions of the promoter to reveal the functional core. I think that the manuscript might be published in the journal provided the authors addressed a series of concerns.
- Page 1, line 34. What about the 5' UTR?
- Page 1, line 36. The word "activate" is misleading here, as it may suggest that the promoter is regulated rather than constitutive.
- Page 2, line 44. What is "fully function pollen"?
- According to Fig. 3, p385 appears to be directly the 5' UTR. Thus, it is uncertain if the pollen-specific expression is due to transcriptional or post-transcriptional effects.
In this regard, what is the transcriptional start site in the construct shown in Fig. 4a?
- Fig. 2c shows that p385 drives the highest expression level in pollen, but also reveals a small significant expression in leaf. However, p847 drives a relatively good expression in pollen with no apparent leakage in other tissues. This is not discussed in the text.
- Why the expression with the 35S-p385 promoter in rice (20.000 units according to Fig. 5b) is much lower than with the minimal p385 promoter (100.000 units according to Fig. 2c)?
- Rather than bar cuttings, the authors should use log scale in the y axis in Figs. 2c, 4c, 5b.
- Please, indicate the statistical test done in Figs. 2c, 4c, 5b.
- Data of Figs. 2c, 4c come from qPCR experiments? Indicate this in the caption.
- The Discussion is too focused on promoter engineering and lacks perspective. I think it could be enlarged with practical applications, alternatives to OsSUT3, etc.
- It is a bit odd to have the Methods section between the Discussion and the Conclusion.
- Page 14, line 260. Please, provide evidence that the expression of this reference gene (b-actin) is constant in the tested tissues.
Author Response
Point 1: Page 1, line 34. What about the 5' UTR?
Response 1: We have added the content on 5’ UTR that as regulatory factor according to the reviewer’s suggestion like below.
This region harbors many cis-elements that influencing the expression of downstream gene, for example 5’ untranslated region (5’ UTR).
Point 2: Page 1, line 36. The word "activate" is misleading here, as it may suggest that the promoter is regulated rather than constitutive.
Response 2: As the reviewer recommends, we changed the word "activate" to "drive".
Point 3: Page 2, line 44. What is "fully function pollen"?
Response 3: We changed " fully function" to " viable" for avoiding confusion like below.
In flowering plants, the proper development of viable pollen is crucial for successful sexual reproduction and maintaining the continuity of a species.
Point 4: According to Fig. 3, p385 appears to be directly the 5' UTR. Thus, it is uncertain if the pollen-specific expression is due to transcriptional or post-transcriptional effects. In this regard, what is the transcriptional start site in the construct shown in Fig. 4a?
Response 4: As mentioned in Fig. 3, p385 is the 5’ UTR of OsSUT3 gene in rice, but it also is part of OsSUT3 promoter. In Fig. 4a, we just regard p385 as the core promoter that could drive pollen-specific expression of target gene. The transcriptional start site (TSS) has not yet been precisely proved in p385::GUS vector, but we will analyze whether or not p385 is the 5’ UTR of GUS gene in constructed vector in the future studies.
Point 5: Fig. 2c shows that p385 drives the highest expression level in pollen, but also reveals a small significant expression in leaf. However, p847 drives a relatively good expression in pollen with no apparent leakage in other tissues. This is not discussed in the text.
Response 5: We have added our discussion in text about the reasons why significant differences of GUS expressions in leaves between p385 and p847 transformants as the following.
Between the two promoters, p385 not only drove the highest expression of GUS in pollen, but also revealed a small significant expression in leaf. However, p847 also drove a relatively good expression in pollen but with no apparent leakage in other tissues. We speculate that the 119-bp region (from -119 to -1) next to ATG contained some cis-elements that could regulate downstream gene expression in leaves. Similar to our finding, GUS pollen-specific activity was not detected in rice transformants carrying p746, which removed the 5’ 183-bp region from p385 (from -385 to -203) (Fig. 3). However, further research is needed to confirm this point.
Point 6: Why the expression with the 35S-p385 promoter in rice (20.000 units according to Fig. 5b) is much lower than with the minimal p385 promoter (100.000 units according to Fig. 2c)?
Response 6: As reviewer mentioned, the expression with 35S-p385 promoter in panicle is much lower than that with the minimal p385 promoter just by comparing the values of GUS expression. However, in general, we do not compare the values from different figures for the reason listed below.
Relative quantification (RQ) values were calculated using the 2−ΔΔCt method which need to choose an expression value as a comparison, such as the expression value of GUS in stem with p847 is a control group in Fig. 2, but the expression value of GUS in stem with 35S-p385 is a control group in Fig. 5. There are different control groups in different figures, so the values from different figures could not be compared.
On the other hand, we reanalysed the expressions of GUS in panicle with p385 and 35S-p385, respectively. The results showed that the expression with p385 was 3.02 times compared with that with 35S-p385. We speculated that the two promoters, 35S and p385, may interfere with each other when fusing them together. However, that should be in-depth studied in future.
We really appreciate reviewer’s in-depth review of our manuscript.
Point 7: Rather than bar cuttings, the authors should use log scale in the y axis in Figs. 2c, 4c, 5b.
Response 7: We have changed to use log scale in the y axis in Figs. 2c, 4c, 5b according to the reviewer’s suggestion. The figures look better than before.
Point 8: Please, indicate the statistical test done in Figs. 2c, 4c, 5b.
Response 8: We have indicated the statistical test done in Figs. 2c, 4c, 5b in captions.
Point 9: Data of Figs. 2c, 4c come from qPCR experiments? Indicate this in the caption.
Response 9: We have added the data of Figs. 2c, 4c come from qPCR experiments in the captions.
Point 10: The Discussion is too focused on promoter engineering and lacks perspective. I think it could be enlarged with practical applications, alternatives to OsSUT3, etc.
Response 10: We have supplemented the practical applications of OsSUT3 promoter in discussion as follows.
In conclusion, p385 was the strongest driver of six tested promoters to determinate the pollen-specific expression of target gene. In addition, p385 also had been proved to have good stability and high activity of pollen-specific expression in heterologous systems. Therefore, it was potential tool in agriculture, especially in male-sterility. For instance, the transgenic male sterile lines and maintainer lines can be generated by introducing a pollen-killer gene, fertility-restoration gene. This method is suitable to most flowering plants for breeding new varieties and increasing production of hybrid seeds.
Point 11: It is a bit odd to have the Methods section between the Discussion and the Conclusion.
Response 11: According to the style demanded by editors of IJMS journals, that was required to insert the Methods section between the Discussion and the Conclusion.
Point 12: Page 14, line 260. Please, provide evidence that the expression of this reference gene (β-actin) is constant in the tested tissues.
Response 12: Before the qRT-PCR, we detected the quantity and concentration of RNA samples by a NanoDrop 1000. On the basis of RNA concentration, first-strand cDNA was synthesized using an equal quantity of RNA and used for qRT-PCR. In this situation, we thought the expression of this reference gene (β-actin) was generally constant in the tested samples.
Reviewer 2 Report
Authors have demonstrated that the p385 promoter, 29 a short and high-activity promoter can drive pollen-specific expression of transgenes in rice and Arabidopsis transformation experiments. In this regard, the experiments were meticulously performed, and the manuscript is well construed with organized tables and figures. After a rigorous review, I could not find any technical and/or scientific issue in the work; however, the language and grammar require minor polishing. Formatting and syntax errors shall be taken care of. The original (unedited) GUS profiling (histochemical localization) images given in Figures 2, 4 and 5 should be provided in the supplementary material.
Author Response
The Certificate of English Language Editing and the original (unedited) GUS profiling (histochemical localization) images are in the attachment.
Point 1: The language and grammar require minor polishing. Formatting and syntax errors shall be taken care of.
Response 1: As the reviewer recommends, we have carried out English language editing service at LetPub and provided the Certificate of English Language Editing.
Point 2: The original (unedited) GUS profiling (histochemical localization) images given in Figures 2, 4 and 5 should be provided in the supplementary material.
Response 2: We have provided the original (unedited) GUS profiling (histochemical localization) images in figures 2, 4 and 5 as supplementary materials.